# 🎹 PIANOMOTION10M: DATASET AND BENCHMARK FOR HAND MOTION GENERATION IN PIANO PERFORMANCE

**Qijun Gan & Song Wang & Shengtao Wu & Jianke Zhu**[*]
Zhejiang University
`{ganqijun,songw,shengtaowu,jkzhu}@zju.edu.cn`

## ABSTRACT

Recently, artificial intelligence techniques for education have been received increasing attentions, while it still remains an open problem to design the effective music instrument instructing systems. Although key presses can be directly derived from sheet music, the transitional movements among key presses require more extensive guidance in piano performance. In this work, we construct a piano-hand motion generation benchmark to guide hand movements and fingerings for piano playing. To this end, we collect an annotated dataset, PianoMotion10M, consisting of 116 hours of piano playing videos from a bird's-eye view with 10 million annotated hand poses. We also introduce a powerful baseline model that generates hand motions from piano audios through a position predictor and a position-guided gesture generator. Furthermore, a series of evaluation metrics are designed to assess the performance of the baseline model, including motion similarity, smoothness, positional accuracy of left and right hands, and overall fidelity of movement distribution. Despite that piano key presses with respect to music scores or audios are already accessible, PianoMotion10M aims to provide guidance on piano fingering for instruction purposes. The source code and dataset can be accessed at `https://github.com/agnJason/PianoMotion10M`.

## 1 INTRODUCTION

The process of learning has been significantly improved with artificial intelligence techniques, which enable individuals to enhance their skills under the guidance of an AI coach (Wang et al., 2019; Zakka et al., 2023; Wang et al., 2024). This can be extended into learning to play the musical instruments. Particularly, piano performance requires a profound understanding of the underlying relationship between musical compositions and their corresponding physical motions. As the rigorous practice and training program are necessary for athletes to master a diverse range of expressive human poses, it entails extensive practices to achieve proficiency in piano fingering and hand movement. To facilitate access to playing guidance, the development of AI piano coach has been spurred.

Large-scale piano-motion datasets are the foundation of a nuanced approach for motion generation, which offer valuable guidance for physical performance and musical expression. The computational challenge of motion generation lies in capturing the nonlinear relationship between musical pieces and the intricate hand motions required for piano playing. The hand poses vary for the same note across different melodies. Moreover, the dynamic nature of musical expression demands a level of continuous motion, which is challenging to be learned from small datasets. These limitations underscore the urgent need for a large-scale piano-motion dataset.

To address the deficiency of the dataset for guiding hand movements and fingerings in playing piano, we introduce a large-scale 3D piano-motion dataset named PianoMotion10M. As shown in Fig. 1, PianoMotion10M contains piano audio tracks, Musical Instrument Digital Interface (MIDI) files, and annotated hand motions with their corresponding videos, meticulously collected from the Internet. It comprises 1,966 pairs of video and music, with a total duration of 116 hours and

---

[*]Corresponding author.

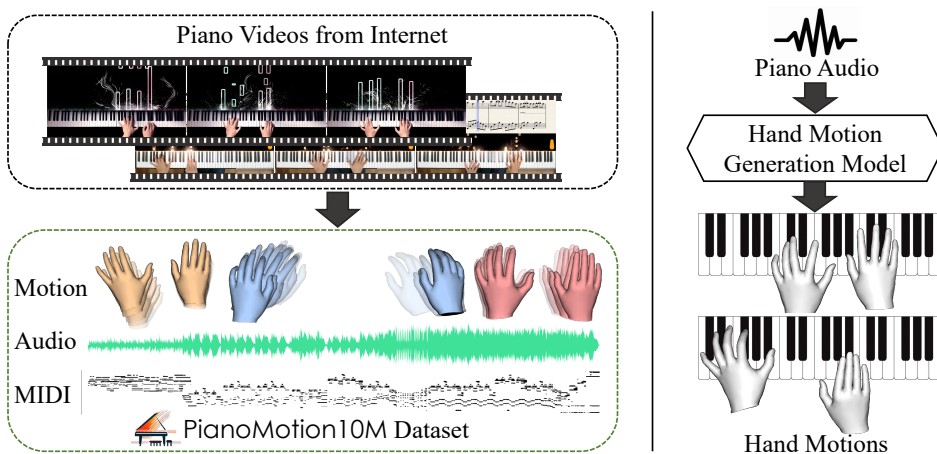

Figure 1: **Overview of our framework.** We collect videos of professional piano performances from the Internet and process them to construct a large-scale dataset, PianoMotion10M, which comprises piano music, MIDI files and hand motions. Building upon this dataset, we establish a benchmark for generating hand motions from piano music.

10 million annotated frames. The parametric MANO hand model (Romero et al., 2017) is employed to represent hand gestures. Our constructed dataset offers a diverse range of music styles and piano techniques, which addresses the demands of various preferences and skill levels.

Traditional applications like PianoPlayer (PianoPlayer, 2018) are adept at generating static hand gestures and positions for piano scores, which typically make use of classifiers to predict the proper fingering combinations. However, they often fail to capture the diversity and continuity inherent of the piano performance in PianoMotion10M. Both rule-based methods (Balliauw et al., 2017; Lin & Liu, 2006) and HMM-based approaches (Yonebayashi et al., 2007; Nakamura & Sagayama, 2015) can estimate fingerings while they primarily focus on the local fingering constraints of continuous notes. Consequently, they often overlook crucial information like long-range fingering relationships, while our task aims to estimate the motions of long clips.

To address these limitations, a novel baseline model is presented to show the effectiveness of PianoMotion10M, which is able to generate realistic hand motions from piano melodies. Given a piece of piano music, our model can locate the positions of both hands and generate a long sequence of hand gestures for the performance. It effectively learns the music-position correlation through an efficient position predictor and produces continuous gestures with a position-guided gesture generator based on a diffusion probabilistic model. To assess our baseline model, we propose several evaluation metrics, including *Fréchet Gesture Distance* and *Wasserstein Gesture Distance* to measure the fidelity of each hand motions, *Fréchet Inception Distance* with a pre-trained auto-encoder to investigate the motion quality of double hands, and *Position Distance* to assess the accuracy of hand positioning, and *Smoothness* of the generated motions.

In summary, our main contributions are: 1) A large-scale piano-motion dataset PianoMotion10M comprises 116 hours of music and 10 million annotated frames with hand poses. To the best of our knowledge, it stands as the first dataset integrating piano music with its corresponding hand motions, which facilitates the tasks of 3D hand motion generation from piano audio tracks and piano music generation conditioned on hand motions. 2) Based on PianoMotion10M, we propose a benchmark with a series of evaluation metrics to investigate the effectiveness on hand motion generation, including the accuracy of positions and fidelity of gestures. 3) A novel baseline model bridges piano music with hand motions, which estimates hand location with a position predictor and generates hand gestures sequences through a position-guided gesture generator.

## 2 RELATED WORK

**Motion-music Datasets.** While multi-modal datasets (Schuhmann et al., 2021; Srinivasan et al., 2021; Miech et al., 2019; Lee et al., 2021) become the key of various learning tasks, there remains a

significant gap in the availability of repository specially designed for music-conditional motion generation. Although the existing hand gesture datasets (Moon et al., 2020b;a; Kwon et al., 2021; Gan et al., 2024; Wu et al., 2023) contain a large number of image-hand gesture pairs, they do not have audio or other related information. This limitation hinders them from the generative tasks, since they mainly focus on hand reconstruction and pose estimation. Recently, datasets like AIST++ (Li et al., 2021) and TikTok (Zhu et al., 2022) are tailored for music-dance learning, which provide limited music segments, typically less than 5 hours in duration. Moryossef *et al.* (Moryossef et al., 2023) manage to automatically detect which fingers pressed the key of the piano and provide a dataset of piano-fingering. However, it does not provide continuous hand gesture movements. Therefore, there is a crucial need to build a piano-motion dataset specially tailored for motion generation tasks. To this end, we introduce the PianoMotion10M dataset, which comprises extensive piano music and their corresponding hand motion annotations.

**Piano Hand Datasets.** Yamamoto *et al.* (Yamamoto et al., 2010) employed a key-frame technique to generate natural hand motions for piano playing from inputted music scores. Liang *et al.* (Liang et al., 2016) used an RGB-D camera to collect 7,200 depth images and designed models to estimate fingertip movements and predict finger tappings. The CMU Panoptic Studio dataset (Joo et al., 2015; 2017; Simon et al., 2017) provided a multi-view dataset of human-object interactions, including annotated hand and face postures. Simon *et al.* (Simon et al., 2017) introduced a real-time hand keypoint detector and a markerless 3D hand motion capture system capable of reconstructing challenging hand-object interactions and musical performances. TwohandsMusic (Seo et al., 2019) presented two shallow networks to estimate 3D hand poses and tap gestures, along with a dataset of 85,000 images of hand movements while playing the piano. To help beginners improve their playing techniques, Johnson *et al.* (Johnson et al., 2020) discussed automatic assessment of pianists' hand postures, which categorizes postures as correct posture, low wrists, or flat hands. Kun *et al.* (Su et al., 2020) collected piano performance videos to generate the music for videos directly. Moryossef *et al.* (Moryossef et al., 2023) managed to automatically detect which fingers pressed the key of the piano and provided a dataset of piano-fingering. Wu *et al.* (Wu et al., 2023) proposed a marker-removal approach for collecting bare-hand data, which included a precise ground truth alongside a large-scale pianist 3D hand dataset, PianoHand2.5M. This dataset provided advanced annotated images for estimating hand postures in marked-hand images.

Our dataset and baseline primarily focus on researching the relationships between piano music and hand postures. More comparisons with other existing hand datasets are shown in the attached pdf document. Since most of existing piano-hand datasets are not publicly available, our open-source release of PianoMotion10M will be of substantial importance to advancing research in this field.

**3D Human Motion Synthesis.** The problem of generating realistic and controllable 3D human motion sequences has been a subject of long-standing study. By taking advantage of 2D keypoint detection (Cao et al., 2017), the synthesis of 2D skeletons has been extensively explored (Ren et al., 2020; Shiratori et al., 2006; Ferreira et al., 2021). Considerable research efforts have been devoted to 2D speech-driven head generation for facial mouth and lip motion generation (Zakharov et al., 2019; Chen et al., 2019; Guo et al., 2021; Ji et al., 2021; Hong et al., 2022), which usually employ either image-driven or voice-driven methods to produce realistic videos of speaking individuals. However, the expressive capabilities of 2D pose skeletons are limited and they are not applicable to 3D character models. Recent methods for full-body 3D dance generation have utilized LSTM (Tang et al., 2018; Yalta et al., 2019; Zhuang et al., 2022), GANs (Sun et al., 2020; Ginosar et al., 2019) or transformer encoders (Li et al., 2021; Huang et al., 2020; Siyao et al., 2022). To generate vivid talking head videos, extensive research has been conducted in the field of speech-driven 3D facial animation (Cao et al., 2005; Cudeiro et al., 2019; Tian et al., 2019; Fan et al., 2022; Zhang et al., 2021). EmoTalk (Peng et al., 2023) animates emotional 3D faces from speech input by generating controllable personal and emotional styles. Tian *et al.* (Tian et al., 2024) introduce a speed controller and a face region controller to enhance stability during the head generation process.

In the domain of hand motion generation, it has been primarily categorized into rule-based methods (Cassell et al., 2001; Huang & Mutlu, 2012; Starke et al., 2022) and data-driven approaches (Kopp et al., 2006; Chiu et al., 2015; Liu et al., 2022; Yoon et al., 2020; Bhattacharya et al., 2021). For instance, Speech2Gesture (Ginosar et al., 2019) utilizes conditional generative adversarial networks to generate personalized 2D keypoints from audio. Ahuja *et al.* (Ahuja et al., 2020) propose a method for personalized motion transfer. Ao *et al.* (Van Den Oord et al., 2017) learn the mapping between the speech and gestures from data using a combined network structure

of the vector quantized variational auto-encoder (VQ-VAE). Beyond 2D keypoints generation, Tri-Modal (Yoon et al., 2020) extracts different upper body movements from TED talks and designs a LSTM-based neural network conditioned on audio, text, and identity to generate co-speech gestures.

Recently, diffusion models have achieved promising results in generating human motions. Previous works such as MDM (Tevet et al., 2022) and MotionDiffuse (Zhang et al., 2024) have produced realistic motion inspired by denoising diffusion probabilistic models (DDPM) (Ho et al., 2020). PhysDiff (Yuan et al., 2023) extends MDM by imposing physical constraints. MLD (Chen et al., 2023) utilizes latent carrier DDPM for forward denoising and reverse diffusion in motion latent space. MAA (Azadi et al., 2023) enhances the performance of non-distributed data by pre-training diffusion models. Zhang *et al.* (Zhang et al., 2023) introduce retrieval-enhanced DDPM, which improves the text-to-motion functionality in distribution.

## 3   PIANOMOTION10M DATASET

It is a formidably challenging task to map piano music to hand motions due to the significant influence of note sequences and positions on hand movements and fingering. Lacking diverse data may lead to the inferior performance on estimating various hand poses in piano playing. To capture the variability, we present the first large-scale piano-motion dataset, PianoMotion10M, which comprises 1,966 piano performance videos along with 10 million hand poses and their corresponding MIDI files, resulting in an overall duration of 116 hours. Fig. 2 presents an example of our dataset. These videos are segmented into 16,739 individual clips with a length of 30 seconds. To ensure comprehensive evaluation, we extract 7,519 clips for training, 821 for validation and 8,399 for testing. The dataset is divided based on the original videos to avoid the overlap in piano performances. The detailed comparisons with the existing datasets on hands and 3D human motions are summarized into Tab. 1.

Table 1: **Comparison between different hand and motion datasets.** The proposed PianoMotion10M dataset consists of piano music with corresponding hand poses for hand motion generation. For reference, the table is organized as follows: the first five rows list existing hand-image datasets, the subsequent four rows present music-motion datasets, and the following four rows display various piano-hand pose datasets. The final row features our dataset.

| Dataset | Year | Pose | Size | Subject | Music | MIDI | Duration(h) | Available |
|---|---|---|---|---|---|---|---|---|
| CMU Panoptic | 2017 | ✓ | 31K | 8 | ✗ | ✗ | - | ✓ |
| FreiHAND | 2019 | ✓ | 134K | 32 | ✗ | ✗ | - | ✓ |
| InterHand2.6M | 2020b | ✓ | 2.6M | 27 | ✗ | ✗ | - | ✓ |
| RGB2Hands | 2020 | ✓ | 1K | 2 | ✗ | ✗ | - | ✓ |
| Re:InterHand | 2023 | ✓ | 1.5M | 10 | ✗ | ✗ | - | ✓ |
| GrooveNet | 2017 | ✓ | - | 1 | ✓ | ✗ | 0.38 | ✓ |
| DanceNet | 2022 | ✓ | - | 2 | ✓ | ✗ | 0.96 | ✗ |
| EA-MUD | 2020 | ✓ | - | - | ✓ | ✗ | 0.35 | ✓ |
| AIST++ | 2021 | ✓ | 10.1M | 30 | ✓ | ✗ | 5.19 | ✓ |
| Barehanded Music | 2016 | ✓ | 7.2K | 2 | ✗ | ✗ | - | ✗ |
| TwohandsMusic | 2019 | ✓ | 85K | 5 | ✓ | ✗ | - | ✗ |
| Piano-fingering | 2023 | ✓ | 155K | - | ✓ | ✓ | - | ✗ |
| PianoHand2.5M | 2023 | ✓ | 2.5M | 21 | ✓ | ✓ | 4.3 | ✗ |
| PianoMotion10M | 2024 | ✓ | 10.5M | 14 | ✓ | ✓ | 116 | ✓ |

### 3.1   DATA COLLECTION

There is a wealth of videos and live streams dedicated to musical instrument performances and tutorials from the Internet. Note that each individual has a unique playing style, we firstly select 14 piano experts from Bilibili[1] and collect a total of 3,647 candidate videos. To ensure consistency and

---
[1] https://www.bilibili.com, one of the most popular video-sharing platforms in China.

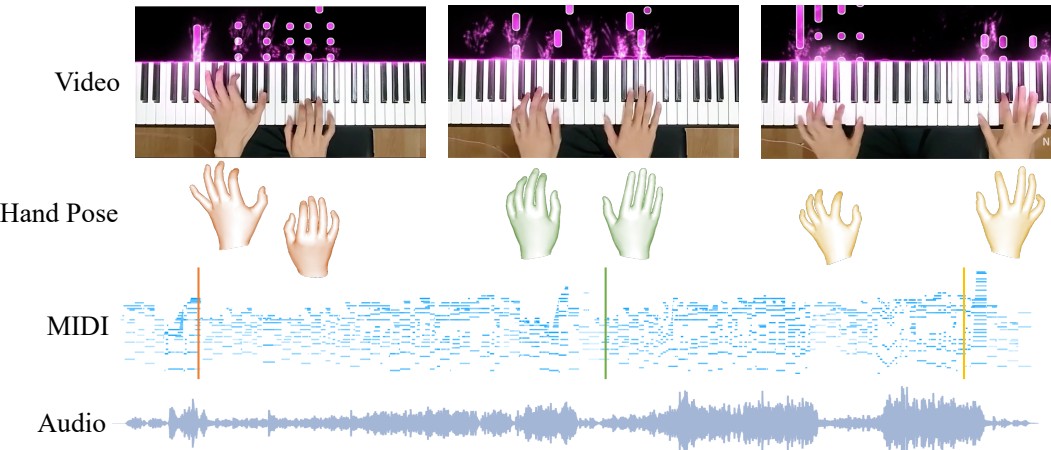

Figure 2: **Illustration of sample from PianoMotion10M.** Each sample in our dataset includes audio, hand pose annotations, and a MIDI file along with the corresponding Bilibili video ID.

enhance the quality of our dataset, we conduct pre-processing with five pivotal factors, including resolution, audio quality, camera perspective, presence of multiple individuals, and visibility of hands. Accurate, noise-free audio is essential to our dataset. We manually select pure piano music to ensure the audio contains no human vocals or sounds from other instruments, rather than employing on automated tools Kong et al. (2020). Moreover, videos with a resolution lower than $1080 \times 1920$ are discarded to ensure the quality of our dataset. To enhance the observation of hand movements and gestures, we choose videos with a bird's-eye view to minimize hand occlusion. Furthermore, we remove those videos where hand regions are frequently (more than 20% frames) obstructed or invisible during performance. Additionally, we do not take consideration of videos with multiple piano players, which is beyond the scope of this dataset. By addressing these factors, the overall quality and coherence of the dataset have been substantially enhanced. Following this preprocessing stage, the result is a collection of $1,966$ high-quality raw videos. Each one showcases the piano playing by an individual, which is captured from a bird's-eye view along with pure piano audio.

## 3.2 DATA ANNOTATIONS

**MIDI** is a universal digital protocol to store musical data for various musical devices, which is served as a digital representation of musical notes, volume, tempo, and other performance parameters. Some creators provide ground-truth MIDI files for their performances. For the remaining piano performances, automatic piano transcription methods (Kong et al., 2021; Maman & Bermano, 2022; Gardner et al., 2021; Toyama et al., 2023; Cheuk et al., 2023) are employed to convert them into MIDI files, while (Kong et al., 2021) demonstrates more stable and robust performance in wild data. To ensure the accuracy of the conversion results, we replay MIDI files and compare them with the original music tracks. Files are adjusted if they exceed thresholds such as timing differences over 30 ms, dynamic variations beyond 10%, or noticeable pitch mismatches that alter melody or harmony.

**Hand Pose** is captured via the parametric hand model MANO (Romero et al., 2017), which serves as our hand prior model. It effectively maps the pose parameter $\theta \in \mathbb{R}^{J \times 3}$ with $J$ per-bone parts and the shape parameter $\rho \in \mathbb{R}^{10}$ onto a template mesh $\bar{\mathcal{M}}$ with vertices $V$. The MANO model enables the simulation of various hand gestures and movements during piano performances, which provides an effective way to study the relationship between hand gestures in playing piano. Due to the non-uniformity of hand sizes and positions in the collected videos, we first employ the MediaPipe hand detection framework (Lugaresi et al., 2019) to obtain bounding box of the hand region. Video frames are cropped according to the detected hand bounding boxes to enhance the robustness of the results. To this end, hand poses in collected videos are annotated using HaMeR (Pavlakos et al., 2023). HaMeR follows a fully transformer-based architecture and reconstructs hand models with increased accuracy and robustness. All images having hands detected by MediaPipe are annotated with hand poses by HaMeR, which result in a dataset of 10 million image-pose pairs.

Table 2: Statistics on the distribution of subjects in the PianoMotion10M dataset, where subject names denote the identity ID of experts.

| Subject Name | Videos | Clips | Time(sec) | Frames | Subject Name | Videos | Clips | Time(sec) | Frames |
|---|---|---|---|---|---|---|---|---|---|
| 1467634 | 337 | 4,359 | 103,293 | 2,237,181 | 66685747 | 535 | 5,136 | 130,352 | 3,520,370 |
| 37367458 | 114 | 859 | 20,802 | 571,525 | 676539782 | 19 | 359 | 11,314 | 187,917 |
| 442401135 | 275 | 1,788 | 52,030 | 1,209,905 | 688183660 | 264 | 872 | 19,074 | 564,699 |
| 478315001 | 285 | 2,674 | 62,615 | 1,792,707 | Others | 137 | 692 | 18,692 | 442,863 |
| **Total Videos: 1,966, Clips: 16,739, Total Duration(hour): 116.16, Annotated Frames: 10,527,167** | | | | | | | | | |

To enhance the smoothness and continuity of our dataset, the generated hand poses require to be cleaned and refined. The results of MediaPipe and HaMeR are usually accurate in most cases, while some inferior results may occur due to rapid motion and image blurring. The Hampel filter (Pearson et al., 2016) with a window size of 20 is utilized to identify these outliers. The outliers and the timestamps where hands are undetected are firstly labeled as missing values. Within a small period, hand movements can be considered as the motion with constant speed. Therefore, the small gap can be interpolated bilaterally. To address these missing data in the time series, missing segments with the frame length $\delta$ less than 30 frames are filled by linear interpolation with respect to their surrounding values. The others are considered as periods when the hand is invisible. To ensure the reliability of detected hands, a similar strategy is employed to label observations with excessively short duration ($\delta < 15$) as invisible. Finally, to ensure the smoothness of hand motions, we make use of a Savitzky-Golay filter (Schafer, 2011) with an order of 3 and a window size of 11 for data smoothing, which delivers better performance in terms of noise separation, artifacts and baseline drifts. The annotated hand poses are manually checked to ensure their quality.

### 3.3 DATA STATISTICS

The dataset is made of contributions from several subjects with different experts, and each provides varying amounts of data in terms of videos, clips, duration, and annotated frames. Tab. 2 presents the detailed statistics on the distribution of subjects within our PianoMotion10M dataset.

The whole piano-motion dataset, PianoMotion10M, consists of 1,966 videos with approximately 116 total hours of footage. Each piece of music is segmented into 30-second clips at 24-second intervals, resulting in a total of 16,739 clips. Note that we discard those clips where hand visibility is below 80%. Our dataset has 14 subjects with different playing styles. This variability ensures a rich diversity of playing techniques and music styles, which is essential for training robust models to predict piano hand gestures from musical pieces. All videos in our dataset are publicly accessible with provided video IDs on the Bilibili website.

## 4 BASELINES

To tackle the challenging task of generating hand motions synchronized with piano audio tracks, we introduce a novel motion generation framework upon the PianoMotion10M dataset. As illustrated in Fig. 3, our framework consists of a position predictor and a gesture generator. The position predictor extracts the hand positions from piano music and integrates them as contextual input for the gesture generator. By leveraging a DDPM model (Ho et al., 2020), the gesture generator estimates hand pose sequence based on the piano audio and the predicted positions. Further details on each component are elaborated in the following subsections.

**Problem Formulation.** Given a piano audio piece $A$ with $N$ frames, our objective is to obtain the hand position sequence $P \in \mathbb{R}^{N \times 3}$ and hand gestures $\Theta \in \mathbb{R}^{N \times J \times 3}$ for both left and right hands.. The hand position sequence $P$ encompasses the 3D coordinates of the left and right hands. The hand gestures $\Theta$ consist of Euler angle at each joint of both hands.

Due to the highly nonlinear relationship between acoustic signals and hand gestures, it poses a significant challenge to estimate motions through discriminative models (Peng et al., 2023), which usually leads to an average pose, as demonstrated in Section 5.2. To address this issue, a hand position predictor is introduced to estimate the continuous changes in hand positions. Moreover, a generative model is utilized to reconstruct hand gestures from a piano music piece. Hereby, the task

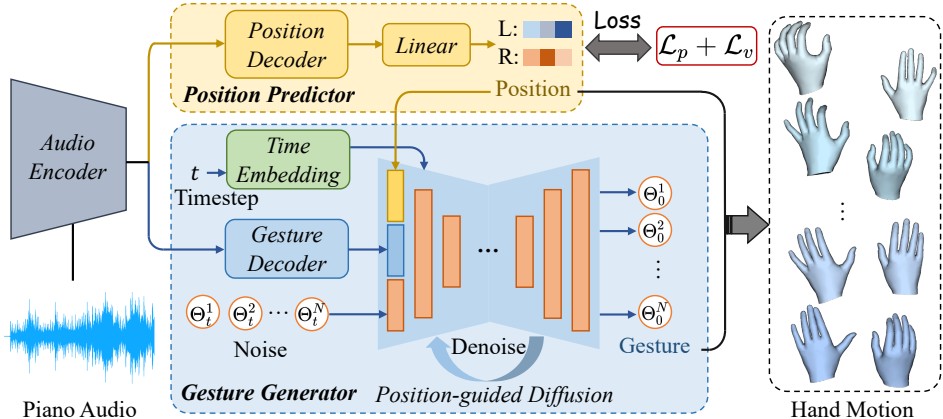

Figure 3: **Illustration of our baseline model.** Given a piece of piano music, our baseline model estimates the hand motions by predicting hand positions and generating hand gestures.

of hand motion generation becomes more concise and comprehensive by disentangling it into hand position estimation and gesture generation. To extract the audio features $f_a$ from the audio $A$, we make use of a pre-trained audio feature extractor $\Phi_a$ (Baevski et al., 2020; Hsu et al., 2021), which is formulated as $f_a = \Phi_a(A), f_a \in \mathbb{R}^{N \times C}$. $C$ is the feature dimension of $\Phi_a$.

## 4.1 POSITION PREDICTOR

The position predictor is employed to predict the 3D positions of the left and right hands, respectively. Since the audio features $f_a$ cannot directly be mapped to the positions, a feature embedding module is treated as our position decoder $\Phi_p$ to extract the latent features. It projects the sequential audio features $f_a$ onto the latent features $f_p$ with more comprehensive temporal information, which is formulated as $f_p = \Phi_p(f_a), f_p \in \mathbb{R}^{N \times 512}$. Subsequently, a linear mapping is employed to project the feature $f_p$ onto the output positions $P$.

In our experiments, we can make use of either Transformer (Vaswani et al., 2017) or State-Space Model (SSM) (Gu & Dao, 2023) as our feature embedding module. Transformer leverages self-attention mechanisms to effectively capture long-range dependencies and contextual information in order to learn temporal relationships. Recently, a different representation inspired by classical SSM (Kalman, 1960) is proposed to replace the attention mechanism, which is built upon a more contemporary Selective Structured State Space Model (S6) (Gu & Dao, 2023) suitable for deep learning. By sharing a similar architecture to the classical RNN, it can efficiently capture information from previous inputs.

## 4.2 POSITION-GUIDED GESTURE GENERATOR

Leveraging recent achievements (Zhi et al., 2023; Azadi et al., 2023) in motion generation, our approach incorporates a diffusion probabilistic model (Ho et al., 2020). By mastering denoising, this model effectively captures the complex distribution of hand motions observed in large-scale piano-motion datasets, which has the capability to generate motions with different conditions. Starting with a clean sample $\Theta_0$ of motion sequence, the forward diffusion process establishes a Markov chain that gradually adds noise to $\Theta_0$ in $T$ steps, which generates a series of noisy samples $\Theta_1, ..., \Theta_T$ as

$$q(\Theta_t|x_0) = \mathcal{N}(\Theta_t; \sqrt{\bar{\alpha}_t}x_0, (1 - \bar{\alpha}_t)I), \tag{1}$$

where $\mathcal{N}$ denotes the Gaussian distribution. $\bar{\alpha}_t = \prod_{s=1}^{t}(1 - \beta_s)$ and $\beta$ represent the variance scheduler for the added noise. Therefore, a model parameterized by a deep neural network $G$ is trained to master the inverse process within another Markov chain, which learns the mapping $p(\Theta_{t-1}|\Theta_t)$ to sequentially denoise samples over $T$ steps. Specifically, denoising model $G$ consists of a 4-layer U-Net (Ronneberger et al., 2015) with 256, 512, 1024, 2048 dimensions for each layer.

To reduce the noise in piano audio, the audio features $f_a$ are simultaneously fed into the denoising neural network $G$ as conditions. Similar to the position decoder $\Phi_p$, a gesture decoder $\Phi_g$ is utilized to extract gesture features $f_g$ from the audio features $f_a$ as $f_g = \Phi_g(f_a)$. Considering that the finger movements at different positions for the same pitch are distinct, $G$ requires the guidance of hand positions $P$ from the position predictor. This will enhance the fidelity of the generation process. As for the additional conditions, the time step embeddings of time $t$ are concatenated in denoising process. The denoising process can be formulated as follows

$$\hat{\Theta}_0 = G(\Theta_t, t; f_g, P), \tag{2}$$

where $\hat{\Theta}_0 \in \mathbb{R}^{N \times J \times 3}$ denotes the result of hand motions within $N$ frames.

## 4.3 Implementation Details

We employ a two-stage scheme to train our proposed network. At the first stage, the position predictor is trained using position error $\mathcal{L}_p$ and velocity loss $\mathcal{L}_v$. The position error $\mathcal{L}_p$ computes the $L_1$ loss between the predicted position $\hat{P}$ and the ground truth $P$, expressed as $\mathcal{L}_p = ||\hat{P} - P||_1$. Inspired by (Peng et al., 2023), velocity loss is adopted to induce temporal stability for generating smoothing movements, which is formulated as $\mathcal{L}_v = ||(\hat{P}_n - \hat{P}_{n-1}) - (P_n - P_{n-1})||_2$. $n$ denotes the $n$-th frame in a motion sequence. Specifically, our model is trained by subject `1467634` and subject `66685747`, which have the similar piano keyboard layout. At the second stage, the parameters of position predictor are frozen, and the estimated positions are employed as a guidance for the gesture generator. As in (Salimans & Ho, 2022), the denoising process is modified from noise prediction to velocity prediction. During the whole training process, the parameters of audio feature extractor $\Phi_a$ are frozen.

The baseline model is implemented with PyTorch. We normalize the inputs into 30 FPS through interpolation, where each piece of music lasts 8 seconds. Both stages involve training for $100,000$ iterations with the learning rates of $2 \times 10^{-5}$ and $5 \times 10^{-5}$, respectively. We conducted all the experiments on a PC with single NVIDIA RTX 3090Ti GPU, which has 24GB of GPU RAM.

## 5 Benchmark

## 5.1 Evaluation Metrics

To assess the performance of our proposed baseline, we employ several evaluation metrics to examine the effectiveness of hand poses generated by the input piano music, which are crucial in understanding the relationship between piano melody and its corresponding playing motions. Specifically, to assess the overall motion distribution similarity of the hand, we employ the Fréchet Inception Distance (FID). In piano performance, the left and right hands execute distinct actions and movements. For a detailed evaluation of the disparity of single-hand movements, we utilize the Fréchet Gesture Distance (FGD) and Wasserstein Gesture Distance (WGD). Evaluating positional accuracy is also crucial, for which we introduce the Positional Distance (PD) as a metric. Smoothness is utilized to determine the fluidity of the generated motions.

**Fréchet Inception Distance (FID).** *FID* is introduced to measure the overall motion similarity by calculating the Fréchet distance between the feature vectors of prediction and ground truth. We pretrain an auto-encoder (Yoon et al., 2020) to project motion sequence onto a latent space for double hands. The FID is adopted to assess the fidelity of the overall motions generated by our baseline model. The formulation of FID is derived as follows

$$FID = ||\mu_{pred}^{\mathbf{f}} - \mu_{gt}^{\mathbf{f}}||^2 + Tr(C_{pred}^{\mathbf{f}} + C_{gt}^{\mathbf{f}} - 2 * \sqrt{C_{pred}^{\mathbf{f}} * C_{gt}^{\mathbf{f}}}), \tag{3}$$

where $\mathbf{f}$ represents the latent features obtained by the autoencoder. $\mu$ and $C$ denote the mean vector and covariance matrix, respectively. $Tr()$ is the trace operation of a matrix.

**Fréchet Gesture Distance (FGD).** On the other hand, *FGD* is utilized to compute the long-term disparity between predicted gestures and ground truth of one hand. It calculates the Fréchet distance over complete sequences of hand motions $\Theta$, offering a metric for the global alignment between generated and real data. The formula is listed as follows

$$FGD = ||\mu_{pred}^{\Theta} - \mu_{gt}^{\Theta}||^2 + Tr(C_{pred}^{\Theta} + C_{gt}^{\Theta} - 2 * \sqrt{C_{pred}^{\Theta} * C_{gt}^{\Theta}}). \tag{4}$$

Table 3: **Quantitative evaluation** of our proposed hand motion generation baseline and existing models on the validation set. We present a comparative analysis of various network architectures, highlighting the performance and efficiency of our baselines in generating hand motion.

| Method | | Decoder | Step | Left Hand | | | | Right Hand | | | | FID↓ | Params (M) |
|---|---|---|---|---|---|---|---|---|---|---|---|---|---|
| | | | | FGD↓ | WGD↓ | PD↓ | Smooth↓ | FGD↓ | WGD↓ | PD↓ | Smooth↓ | | |
| EmoTalk (2023) | | TF | - | 0.445 | 0.232 | 0.044 | 0.353 | 0.360 | 0.259 | 0.033 | 0.313 | 4.645 | 308 |
| LivelySpeaker (2023) | | TF | 1000 | 0.538 | 0.220 | 0.038 | 0.406 | 0.535 | 0.249 | 0.030 | 0.334 | 4.157 | 321 |
| Our-Base | Wav2vec | SSM | 1000 | 0.425 | 0.223 | 0.042 | 0.412 | 0.416 | 0.246 | 0.034 | 0.335 | 3.587 | 320 |
| | | TF | 1000 | 0.426 | 0.219 | 0.040 | 0.402 | 0.424 | 0.246 | 0.033 | 0.334 | 3.608 | 323 |
| | HuBERT | SSM | 1000 | 0.432 | 0.218 | 0.041 | 0.407 | 0.402 | 0.247 | 0.033 | 0.336 | 3.412 | 320 |
| | | TF | 1000 | 0.432 | 0.219 | 0.041 | 0.412 | 0.418 | 0.247 | 0.034 | 0.338 | 3.529 | 323 |
| Our-Large | Wav2vec | SSM | 1000 | 0.430 | 0.219 | 0.038 | 0.253 | 0.421 | 0.244 | 0.031 | 0.208 | 3.453 | 539 |
| | | TF | 1000 | 0.372 | 0.214 | 0.038 | 0.251 | 0.354 | 0.244 | 0.030 | 0.209 | 3.376 | 557 |
| | HuBERT | SSM | 1000 | 0.406 | 0.217 | 0.037 | 0.237 | 0.403 | 0.244 | 0.030 | 0.214 | 3.395 | 539 |
| | | TF | 1000 | 0.372 | 0.217 | 0.037 | 0.248 | 0.351 | 0.244 | 0.030 | 0.205 | 3.281 | 557 |

**Wasserstein Gesture Distance (WGD).** *WGD* (Rubner et al., 2000) is computed between two distributions, each of which is represented as a Gaussian Mixture Model (GMM) (Kolouri et al., 2018) as below

$$W(\mathcal{I}_x, \mathcal{I}_y) = \inf_{\gamma \sim \Pi(\mathcal{I}_x, \mathcal{I}_y)} \mathbb{E}_{(x,y) \sim \gamma} \left[ \|x - y\| \right], \tag{5}$$

where $\Pi(\mathcal{I}_x, \mathcal{I}_y)$ indicates the joint distributions that combine the parametric GMM distributions $\mathcal{I}_x$ and $\mathcal{I}_y$. Distributions $\mathcal{I}_x$ and $\mathcal{I}_y$ are fitted by predicted gestures and ground truth of single hand. $\mathbb{E}$ calculates the expectation of the sample pairs $(x, y)$ with multiple Gaussians, which focuses on short-term similarity by analyzing mixture distributions, allowing it to capture finer, localized motion patterns within the sequence. The WGD metric offers a robust measure of dissimilarity between the generated motion and ground truth.

**Position Distance (PD).** The *PD* calculates the $L_2$ distance between estimated positions and the ground truth, which is formulated as

$$PD = \|P_{pred} - P_{gt}\|_2^2, \tag{6}$$

where $P$ is the hand root position. It is essential to evaluate the precision of predicted hand positions to ensure the accuracy required for piano fingerings.

**Smoothness.** *Smoothness* is measured by computing the acceleration of each joint. However, hands in a static state exhibit maximum smoothness, which are contrary to the desired outcome. Consequently, we consider the acceleration of ground truth as a reference and utilize relative acceleration as the evaluation metric for smoothness, which is formulated as $Smooth = \sum_i |\hat{\tau}_i - \tau_i|$. $\tau_i$ denotes the acceleration of the $i$-th joint. It reflects the continuance and coherence of the estimated gestures.

## 5.2 EXPERIMENTS

**Experimental Setup.** Within the network, we compare two transformer-based audio feature extractors, Wav2Vec2.0 (Baevski et al., 2020) and HuBERT (Hsu et al., 2021), which are popular in the field of self-supervised speech recognition. Regarding the feature embedding module (Decoder), we evaluate the performance of SSM-based model (Gu & Dao, 2023) in contrast to the transformer (TF) approach (Vaswani et al., 2017). We conduct experiments using two different model configurations. Both of them employ the same diffusion architecture while differing in the feature extractor setup. The base model incorporates Wav2Vec2.0-base/HuBERT-base as the audio feature extractors, featuring a model dimension of 768 and containing 8-layer TF/SSM decoder. The large model utilizes Wav2Vec2.0-large/HuBERT-large, with a model dimension of 1,024 and 16-layer TF/SSM decoder.

**Quantitative Results.** Tab. 3 provides the results of existing methods and our baseline under various experimental settings on PianoMotion10M dataset. Due to the absence of prior work on generating gestures for piano music, we refer to the network structures of EmoTalk (Peng et al., 2023) and LivelySpeaker (Zhi et al., 2023) and re-implement them to account for our task. EmoTalk directly generates poses from audio, while LivelySpeaker utilizes an MLP-based diffusion backbone for gesture generation. Our baseline models achieve better fidelity on hand motions by estimating positions and gestures, separately. In our baseline models, TF-based models outperform the SSM-based models in processing our time-series information, especially in our large model. For the audio feature extractor, the performance of the HuBERT model slightly outperforms the Wav2vec2.

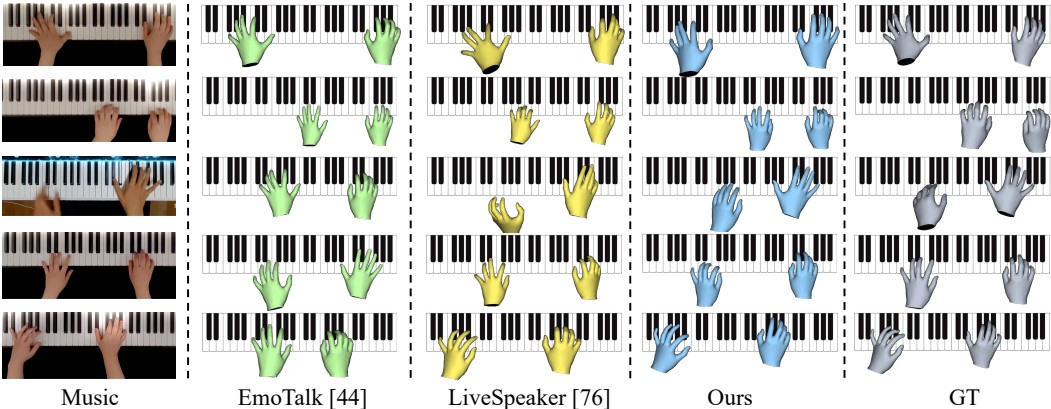

Figure 4: **Illustration of the qualitative results.** We display the generated gestures across frames using different methods. Our method stands out due to its greater fidelity, as shown in the examples.

Additionally, we conduct ablation experiments on different denoising steps and our model also achieves competitive results with fewer steps, as shown in Table 4.

**Qualitative Results.** Fig. 4 demonstrates the visual results of our baseline and the existing models. The output of EmoTalk method exhibits a static average gesture. While MLPs afford LivelySpeaker rapid inference speed, they compromise the fidelity of generated motions.

| Step | FID↓ | FGD↓ | WGD↓ | Smooth↓ |
|------|-------|-------|-------|---------|
| 5 | 3.540 | 0.361 | 0.237 | 0.354 |
| 10 | 3.682 | 0.363 | 0.236 | 0.310 |
| 100 | 3.438 | 0.366 | 0.232 | 0.254 |
| 300 | 3.360 | 0.348 | 0.233 | 0.240 |
| 1000 | 3.281 | 0.362 | 0.231 | 0.227 |

Table 4: **Ablation study** on denoising steps.

Conversely, our model demonstrates notably superior performance compared to previous methods by taking advantage of a two-stage approach, as illustrated in Fig. 4. To attain accurate positional information, we utilize an end-to-end position predictor rather than making use of an uncontrollable diffusion model for position generation. Furthermore, we employ a position-guided approach with a diffusion-based gesture generator for hand motion estimation.

## 6 CONCLUSION

In this work, we present PianoMotion10M, a new large-scale piano-motion dataset for hands, which has 116 hours of piano music and 10 million frames annotated with hand poses. We address the critical issue that the current datasets are insufficient for human-piano interaction. Based on PianoMotion10M, we develop a benchmark model that maps the piano music pieces to hand motions. To simplify the learning target, we divide the motion into position and gesture by introducing a position predictor along with a gesture generator guided by the estimated positions. We suggest the evaluation metrics to measure the fidelity and smoothness of the hand motions compared to the ground truth. Our dataset and benchmark will further advance the automation of piano performance simulation and assist in learning piano playing.

## 7 ACKNOWLEDGMENTS

This work was supported in part by the National Key Research and Development Program of China under Grant (2023YFF0905104) and in part by the National Natural Science Foundation of China under Grants (62376244). It is also supported by the Information Technology Center and State Key Lab of CAD&CG, Zhejiang University. We are grateful to all Bilibili creators who contributed to the dataset construction, especially *CIPMusic*, for their support.

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

APPENDIX

In this part, we further provide more details and discussions on our proposed dataset and benchmark:

- §A: More statistics of PianoMotion10M;
- §B: More visual samples of PianoMotion10M dataset;
- §C: More details of our baseline;
- §D: Limitation and future work;
- §E: Ethical considerations;
- §F: Author statement;

## A    MORE STATISTICS OF PIANOMOTION10M

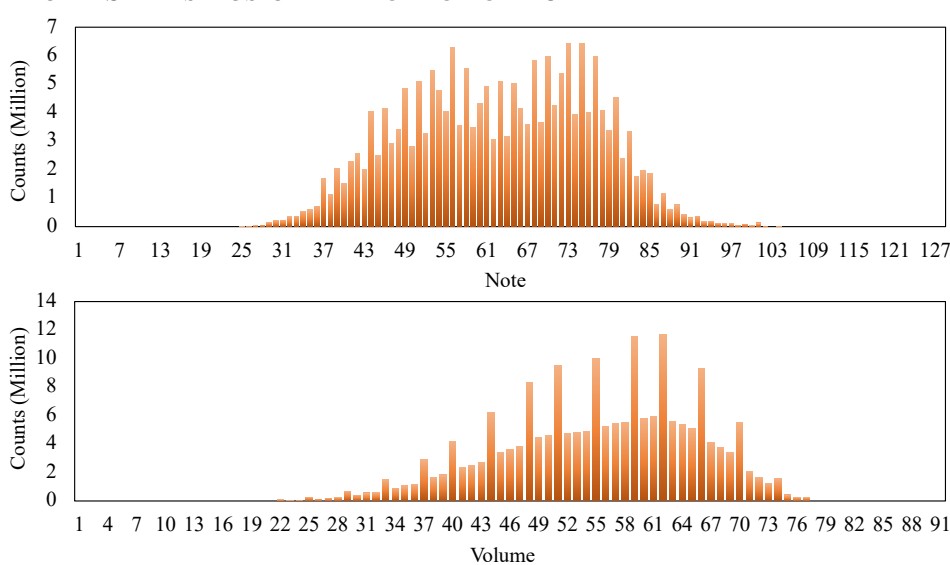

Figure A1: **Distribution of Note Clicks and Volume Levels in the PianoMotion10M Dataset.** The top figure depicts note click frequency, and the bottom one shows the volume distribution.

Fig. A1 presents a detailed statistical analysis of piano fingerings, which focuses on the frequency of note clicks and the distribution of volume levels.

**Note Click Counts.** The top figure in Fig. A1 displays the frequency of each note played, measured in millions. This distribution spans 128 keys of the piano, which indicates frequent usage of those keys in performances. It shows higher counts in specific note ranges.

Notes around the middle of the keyboard, particularly from C4 to C6, exhibit significantly higher click counts, aligning with their common use in melodies and harmonic accompaniments. Conversely, notes in the low (A0 to B1) and high (C7 to C8) octaves have markedly fewer clicks, as these ranges are less frequently used and typically reserved for specific musical effects or embellishments. It is worth noting that, certain notes, particularly those fundamental to common chords and scales (e.g., A, C, E, and G), exhibit higher frequencies, reflecting their frequent use in various musical pieces.

**Volume Distribution.** In addition to note frequency, the figure below presents a comprehensive distribution of volume levels, spanning various ranges to highlight the dynamics of piano playing. Volume counts, measured in millions, provide insights into the intensity and expression captured in our constructed dataset.

There is a higher count of notes played at moderate volume levels. This reflects the natural dynamics of piano playing, where most notes are neither extremely soft nor loud.

## B    MORE VISUAL SAMPLES OF PIANOMOTION10M DATASET

We provide more visual samples of PianoMotion10M dataset, as illustrated in Fig. A2.

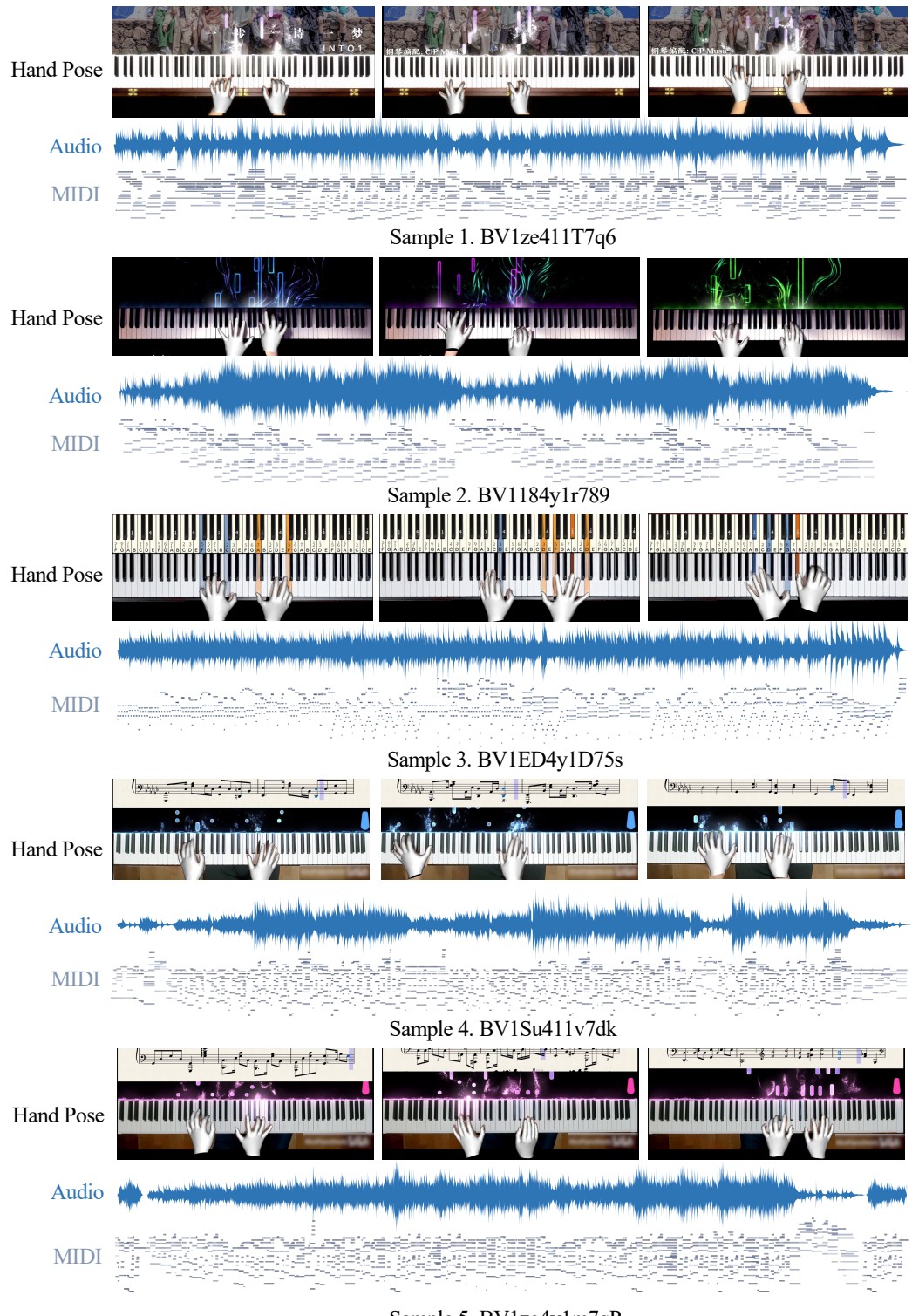

Figure A2: **More samples from PianoMotion10M dataset.** BV*** denote the corresponding video iDs.

## C  MORE DETAILS OF OUR BASELINE

The audio feature extractor $\Phi_a$ maps the audio $A$ to the feature vector $f_a$. We use pre-trained Wav2Vec2.0 (Baevski et al., 2020) and HuBERT (Hsu et al., 2021) models developed by Facebook AI for $\Phi_a$. Both models leverage extensive unlabeled data for unsupervised pretraining to learn high-dimensional speech representations. HuBERT extends its semi-supervised learning approach with pseudo-labels to self-supervised learning. In our experiments, we use `wav2vec2-base-960h` and `hubert-base-ls960` for the base model, and `wav2vec2-large-960h-lv60-self` and `hubert-large-ls960-ft` for the large model as the audio feature extractor $\Phi_a$.

## D  LIMITATION AND FUTURE WORK

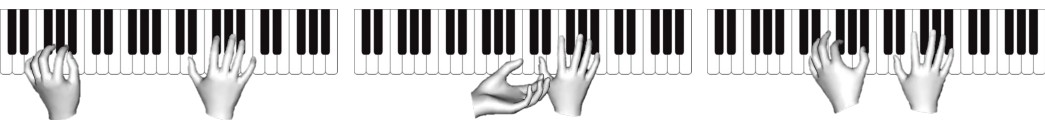

Figure A3: Samples of failure in the generated hand motions when dealing with excessively fast music.

Our dataset is derived from 14 subjects and faces an issue of imbalance, primarily due to the varying levels of performance provided by authorized creators. In extreme performance scenarios, such as rapid, multi-hand, or non-human piano audio, our baseline method may yield incorrect hand transitions due to out-of-distribution challenges, as illustrated in Fig. A3. Additionally, not all videos meet the required data quality standards. In the future, we plan to engage creators on other platforms, such as YouTube and TikTok, to further enhance the diversity of our dataset.

Our dataset closely associates piano music with hand movements. Due to data diversity, we have not yet aligned piano positions in all videos. We plan to engage extra experts to annotate piano key positions for more precise spatial alignment in the next version. Additionally, the variance in piano tones across recordings may also affect the baseline model's performance.

PianoMotion10M provides piano music, corresponding MIDI files, and hand poses, offering researchers a valuable resource for studying human-piano interaction. This dataset enables the analysis of piano music through hand gestures and the generation of hand poses from audio tracks. With PianoMotion10M, we hope to benefit and facilitate further research in relevant fields.

## E  ETHICAL CONSIDERATIONS

Piano motion datasets may pose significant privacy challenges, particularly concerning the pianist's identifiable aspect, mainly their hands, during piano performance. Our dataset comprises videos uploaded by users on Bilibili, which are publicly accessible. During the dataset collection process, we carefully selected the high-quality piano performance videos with a bird's-eye view on Bilibili and obtained permissions from the respective creators to use their videos, and accessed the original content. Those videos from creators who did not grant permission or did not respond have not been included in the dataset. We informed the creators that we would extract hand poses and audios from the videos to create a publicly available, non-commercial dataset. We clarified that their personal likenesses would not be used. The dataset also included metadata, such as the video URL, to trace the original video. In the camera-ready version, we will acknowledge all the creators' contribution in building our dataset.

## F  AUTHOR STATEMENT

The authors bear all responsibility in case of violation of rights. We confirm that the PianoMotion10M dataset is open-sourced under the CC BY-NC 4.0 International license and the released code is publicly available under the Apache-2.0 license, ensuring open access and permissive usage for academic and research purposes.

