# OpenReview forum: "PianoMotion10M: Dataset and Benchmark for Hand Motion Generation in Piano Performance"
_ICLR.cc/2025/Conference — ICLR 2025 Spotlight_

### Official Review · Reviewer_ik5L · 2024-10-31

**Soundness:** 3
**Presentation:** 3
**Contribution:** 3
**Rating:** 8
**Confidence:** 4

**Summary:**

This paper proposes a large dataset consisted of piano music and aligned MIDI and hand positions containing 116h of recordings and 14 subjects. The goal of the dataset is to train generative processes to renderize 3d human-hand motion videos. This work presents an interesting baseline method to renderize hands.

**Strengths:**

In descending order of importance:

- A large dataset for hand pose generation is proposed. This is original, clear, and significant.
- A model is proposed. This is technically original, but the significance is not well discussed for the reasons stated in the comments.
- A set of measures for performance is proposed. They are not original, nor their significance to the problem is discussed.

Overall, I find the article interesting, although I am not sure how many people are specifically trying renderize hand poses in pianos using audio as input. However, this dataset could also be used for other tasks, which broadens its impact.

- The initiative of making a dataset for this task is great, and the ethics statement covers most of the potential problems,
- The dataset has some potential problems that could be taken care of, especially the long-term availability and the reliance on a specific company and several users to maintain the data,

**Weaknesses:**

There is a great emphasis in the proposed model whereas the proposed dataset and measures are not well discussed. I would suggest focusing the paper on the dataset and on the discussion on the metrics and skip the model proposal altogether. In special, this is because the model uses more data than any other model, hence it is expected to approach the data distribution better than other models.

- The presented model is interesting, but its performance is close to EmoTalk (more on that in item 4)
- I miss some discussions on the metrics and their relationship with the problem, and the space taken by describing the baseline method could be used for that. In fact, EmoTalk and LivelySpeaker, according to Table 3, are already interesting for future comparisons and their metrics are very close to the proposed method's.

One aspect that has not been addressed is a description of styles, genres, levels of expertise, and other biases in the dataset. This leads to thinking that the dataset polarized towards particular styles of playing (for example: ragtime versus classical). This is not a problem per se, as it is unavoidable; however, not *knowing* what these biases are can become a problem.

**Questions:**

I do have questions regarding the text:

Section 4.3: The learning rates are said to be 2e^{-5} and 5e^{-5}. The math reads: 2 (or five) times exp(-5). The usual reporting method is to use 2 (or 5) times 10^{-5}. Was that a typo, or are you really using the Euler number to define learning rates? If so, is there a reason for such?

Section 3.2:
- MIDI files with high discrepancies were adjusted: what is the measure for a "high discrepancy"? The issue here is that, if this dataset is going to be used as a benchmark, there should be a measure on how much it differs from reality.

- Data uses a Savitzky-Golay filter. This filter is applied to the data in the dataset to increase smoothness. What are the filter parameters (order, frequency response, etc.)? Also, why was this specific type of filter chosen instead of the standard ones (a Butterworth filter seems adequate)? What is the impact of not using the filter?

Section 3.3: All videos are accessible with the IDs in the Bilibili website. Is Bilibili aware and consenting with the idea that there will probably be robots scraping their data without using their website? That is, are there chances that Bilibili will simply include barriers that would harm the dataset? Also, if users take down their own videos, how would that impact the dataset?

Section 4.3: The GPU has 24GB of RAM. How big is the model itself? How long did training take? Also, are you planning on releasing the trained weights?

Section 5.1:
- Several metrics are proposed, which is a good thing. However, none of the metrics were discussed in terms of their relevance to the real problem. For example: what kind of "error" in the rendering could be highlighted by the FID, FGD or WGD? Without this discussion, the metrics simply pose a "competition" on who gets better values. A phrase such as "The WGD metric offers a robust measure of dissimilarity between the generated motion and ground truth." does not clarify if this dissimilarity consists of a jittery behavior, an offset, an error in velocity, fingers being rendered too long (or too many!), and so on. Such a discussion, with clear examples, could clarify why these specific metrics were used, and not others.
- Also, the metrics imply in a comparison with a ground truth. This is not wrong per se, but, when hands are rendered, is there really a "ground truth"? That is, if two pianists play the same excerpt, wouldn't their hands be slightly different? So, more than increasing the proximity to a ground truth, the goal of the algorithms should be to obtain results that are within a distribution for human differences, not simply to approach a unique ground truth.
- In this same line of thought, does it really make a visible difference to have small variations in these metrics? For example, going from FGD=0.360 to FGD=0.351 is a difference of 0.009 - what type of difference does that reflect in the actual performance?
- How do these measures compare with a Mean Opinion Score (MOS)? It would be great to have some evidence that the proposed measures can substitute MOS, because collecting human opinions is costly. However, we need evidence for such.
- Minor suggestion: in Table 3, make the Right Hand appear to the right of Left Hand.
- A not so minor aspect: why are the measurements so different between the hands? Afterall, we have the same amount of data and the same models for them, correct? According to the data, Wav2Vec+TF is better at rendering left hands, while HuBERT+TF is better at right hands (according to WGD), which seems to indicate that their performance is actually the same and the differences are due to noise/chance/variance.

Section 5.2
- Figure 4 states that "Ours" is close to "GT" (or: the blue hands are closer to the gray ones). This is true, at least for this example; but in general the green and yellow hands seem to be equally plausible hand positions. In the third line, the yellow hands (LiveSpeaker) seem to be even more emphatic on the fact that the left hand is not playing anything.

---

> ### Author Response · Authors · 2024-11-20
> **Response to Reviewer ik5L (Part 1/2)**
>
> ### Q1: Emphasis on the Dataset and Evaluation Metrics
>
> **R:** Thanks for your suggestion. We have enhanced **our description of the dataset and measures**. As we recognize the importance of focusing on the dataset and metrics, we believe that including the proposed model serves as a practical demonstration of the dataset’s reliability and potential applications. The model establishes an initial benchmark for future works, offering a foundation for comparative studies. It demonstrates the potential of the dataset to enable learning of complex, high-quality hand motions that are synchronized with music.
> As mentioned in comments, the comparison results with EmoTalk and LivelySpeaker show that these baseline methods have close performance.
> We highlight that these methods are served as valuable references, the design of our model along with our collected dataset, showcases its scalability for performance improvements and reveals insights into hand motion generation’s unique demands in music contexts. In terms of the evaluation metrics, the additional details are provided in Q6.
>
> ### Q2: Dataset Bias and Style Representation
>
> **R:** Thank you for bringing this to our attention. Our dataset currently encompasses a variety of musical styles, including classical and modern pieces, while specific styles and genres haven’t yet been labeled or quantified. This may introduce certain biases.
> In future work, we plan to annotate and categorize each music piece by **title** and **style**, like the MAESTRO dataset. While this process will require significant time and resources, we consider PianoMotion10M as an evolving project. Moving forward, we plan to reach out to creators on other video sharing platforms, such as YouTube and TikTok, to further enhance the diversity of our dataset.
>
>
> ### Q3: Mistype of Learning Rates
>
> **R:** That was indeed a typo. We are using **standard notation**, with learning rates of $2 \times 10^{-5}$ and $5 \times 10^{-5}$. The text has already been corrected accordingly.
>
>
> ### Q4: More Details of Dataset Process
>
> **R:** Thanks for your question. As mentioned in the R2 of **Global Author Rebuttal**, we have incorporated more details of dataset process and revised the manuscipt accordingly.
>
> ### Q5: More Details of Implementation
>
> **R:** Our large HuBERT-Transformer model has approximately **557 million parameters**, as described in Table 3 of the manuscript. The training process took around **1 day** on a GPU with 24GB RAM. Regarding the release of the trained weights, the checkpoint for our base model is available at [PianoMotion10M](https://github.com/PianoMotion10M/PianoMotion10M/blob/main/logs/diffusion_hubertbase_tf.zip). We intend to release the larger pretrained checkpoints on Hugging Face upon the paper's acceptance.
>
> ### Q6: More Details of Evaluation Metrics
>
> **R:** Thanks for your insightful comments. We understand the importance of clarifying the relevance of the metrics to the real problem and addressing the concerns regarding the comparison with the ground truth. To make it clear, we compare the generated **hand poses** with the ground truth hand poses with joint Euler angles, not the rendered hands. The metrics we used are designed to evaluate different aspects of the generated motion in relation to the true hand poses. Since directly comparing the generated data with the ground truth is unreasonable, we primarily use distributional difference metrics to evaluate our generated results. In piano performance, **the left and right hands perform distinct actions**, requiring separate evaluations.
>
>
> - **FID (Fréchet Inception Distance)** is used to evaluate the overall similarity between the distributions of generated motions and the ground truth, helping to assess the quality of the generated motion sequences at a high level.
> - **FGD (Fréchet Ground Distance)** and **WGD (Wasserstein Ground Distance)** focus on the similarity of **individual hand movements**, specifically assessing how well the generated hand poses match the ground truth in terms of the overall motion characteristics. FGD evaluates long-term distribution similarity across entire sequences, providing a measure of global alignment between generated and real data. In contrast, WGD is based on parametric GMM (Gaussian Mixture Model) distributions, which focuses on short-term similarity by analyzing smaller time segments, allowing it to capture finer and localized motion patterns within the sequence.
> - **PD (Position Distance)** is used to evaluate the **accuracy** of the hand poses at specific locations (e.g., joint positions), ensuring the generated hand poses are spatially accurate. A higher position distance denotes larger errors of the hand position.
> - **Smoothness** assesses the **continuity and smoothness** of the generated hand motion, highlighting if the motion sequence has unnatural jumps or jitter. A higher smoothness value indicates greater jittering in the generated motion.

---

> ### Author Response · Authors · 2024-11-20
> **Response to Reviewer ik5L (Part 2/2)**
>
> ### Q7: Issue of Ground Truth
>
> **R:** We acknowledge that hand motions can exhibit notable variability among pianists, even when playing the same passage.
> Despite individual nuances, these movements typically align with established principles of piano playing technique.
> Instead of attempting to strictly mimic an individual’s motions, our approach leverages a generative model to learn **generalized hand movements** that encapsulate the essence of piano playing techniques.
> This enables the method to produce results within a distribution of natural human variation, aligning with the diverse range of human movement patterns.
>
>
> ### Q8: Difference of Minor Quantitative Variation
>
> **R:** A minor quantitative variation, such as a shift from FGD=0.360 to FGD=0.351, suggests a high similarity between the distributions of generated and real motions. It could reflect subtle differences in finer motions, such as minor finger tremors or slight adjustments, which may still impact the perceived naturalness in detailed movements. Minor metric variations observed may stem from noise, random fluctuations, or subtle hand-specific movement discrepancies, rather than intrinsic differences in model performance.
>
>
> ### Q9: User Study
>
> **R:** In the R5 of **Global Author Rebuttal**, a small range of Mean Opinion Score (MOS) is presented.
> The human assessment results in the table align closely with our proposed benchmark metrics. For example, in _rhythmic synchronization_, EmoTalk shows the lowest performance, while our Transformer-based method performs the best—consistent with the _FID_ and _WGD_ scores in the quantitative evaluations, validating these metrics’ ability to assess short-term action coherence. For _smoothness_, the Transformer-based model is slightly less effective than the Mamba-based model, aligning with the _Smooth_ metric. In _motion diversity_, the Transformer-based method scores significantly higher than others, aligning with _FGD_ and _FID_ scores and supporting their effectiveness in evaluating overall hand motion distribution.
>
> ### Q10: The layout of Table 3
>
> **R:** Thanks for your advice. We have reorgnized Table 3 to place the Right Hand to the right of the Left Hand accordingly.
>
>
> ### Q11: Interpretation of Qualitative Results
>
> **R:** In Figure 4, the results of EmoTalk and LiveSpeaker exhibit some positional differences. In particular, the yellow hands (LiveSpeaker) show a clearer distinction, which emphasizes a positional bias in the simulation.
>
> While variations in hand positions are expected, especially for tasks like piano playing, even small deviations in finger and hand positions are important for maintaining realistic motion. By refining the movement to better match real-life performances, we can achieve more accurate and convincing simulation of human actions.
>
> ### Q12: Data Availability and Compliance Concern
>
> **R:** Thanks for your comments. Please refer to our previous response (R3) provided in **Global Author Rebuttal**. All data in our dataset has been obtained with explicit consent from the creators, who provided their original videos directly. If creators choose to retract their videos from public view, we are committed to honoring their wishes and will remove the data from our dataset as needed.

---

> > ### Comment · Reviewer_ik5L · 2024-11-26
> >
> > Thanks. These reviews address my concerns.

---

> > > ### Author Response · Authors · 2024-11-26
> > >
> > > We are sincerely grateful that our rebuttal addressed your concerns, and we truly appreciate your more positive evaluation of our work (**6 → 8**). Thank you for your supportive and thoughtful feedback!

---

### Official Review · Reviewer_wnCC · 2024-11-01

**Soundness:** 3
**Presentation:** 3
**Contribution:** 3
**Rating:** 6
**Confidence:** 5

**Summary:**

The paper addresses the challenge of hand motion generation for piano performance, recognizing the gap in current AI systems for musical instruction, particularly for instruments like the piano where hand positioning is critical.  The proposed PianoMotion10M serves as an important dataset aimed at solving this problem.

**Strengths:**

1. This is a large-scale dataset comprising 116 hours of piano performance from 14 professional players and 10 million annotated frames. The videos are collected under specific constraints (e.g., high resolution, bird’s-eye perspective) to ensure quality and consistency.

2.  The data collection and annotation method is well-structured, using tools like MediaPipe and HaMeR for robust hand pose detection.

3. The dataset is indeed a valuable contribution, likely to support advancements in both piano instruction and broader motion generation research.

4. The two-stage design—featuring a position predictor and a position-guided gesture generator—seems well-suited for this task and shows reasonable quality. The model uses a diffusion-based architecture with a U-Net structure, which allows it to generate natural, fluid gestures that are better suited to piano performance.

**Weaknesses:**

1. The paper acknowledges the dataset’s limitations, particularly the need for improved diversity in piano positions and video sources, now only include 14 people. Addressing the inconsistency in piano tones and alignment of keyboard positions remains critical for achieving high-fidelity results.

2. A suggestion to include related work: Audeo: Audio Generation for a Silent Performance Video. Kun et,al (NeurIPS 2020).

**Questions:**

1. Do authors have some human evaluation results for the generated motions? It would be better if these could be incorporated.

2. Ethic concern and copyright issue: Since permissions were obtained from Bilibili users, how does the dataset’s open accessibility align with potential copyright restrictions or privacy concerns, particularly for identifiable hand features? I know that they do have some restrictions on this.

**Details Of Ethics Concerns:**

See questions 2 above.

---

> ### Author Response · Authors · 2024-11-20
> **Response to Reviewer wnCC**
>
> ### Q1: Human Evaluation Results for Generated Motions
>
> **R:** Thank you for the suggestion. We agree that human evaluation can provide valuable insights into the realism and quality of the generated motions. In **Global Author Rebuttal** (R5), we provided a small-scale user study in various aspects of generating action quality. Since we currently focus on quantitative metrics, we are planning to conduct more extensive human evaluations to further validate the effectiveness of our model. These results will be incorporated into future versions to enhance our evaluation and provide a more comprehensive assessment of motion realism.
>
> ### Q2: Ethical Concerns and Copyright Considerations
>
> **R:** Please refer to our previous response (R3) in **Global Author Rebuttal**. Our dataset mitigates privacy and copyright concerns by exclusively comprising hand pose data, corresponding audio, and MIDI files, thereby excluding images that could disclose identifiable personal features. This avoids privacy issues related to facial or other identifiable characteristics and adheres to copyright restrictions by not directly distributing visual content.
>
> ### Q3: Missing Reference
>
> **R:** Thanks for your recommendation. We have **cited** the referenced paper in the revision accordingly.

---

### Official Review · Reviewer_Jc4e · 2024-11-02

**Soundness:** 2
**Presentation:** 3
**Contribution:** 3
**Rating:** 6
**Confidence:** 3

**Summary:**

The paper has two main contributions:
1) A dataset of 100+ hours of piano performance audio and (overhead) video, MIDI, and hand motion sequences, all aligned.
2) A baseline method for generating hand motion sequences from piano audio.

**Strengths:**

The main strength of the paper is that this is a really cool dataset linking piano performance audio (and MIDI) to hand motion sequences.  Piano playing is a challenging task requiring extreme dexterity, and this dataset seems like an important contribution to the research community.

The paper is clearly written and provides something new and of value.

**Weaknesses:**

The biggest thing missing is some sort of verification that a hand motion sequence *actually results in the target audio* when applied to a piano, using a physical model.  I recognize that this may be considered out of scope, but without it it's hard to trust even the ground truth provided in the dataset.  And then all of the metrics for the baseline method end up comparing generated hand motions with the ostensible ground truth hand motions, which has two issues:
1) The ground truth hand motions may have errors.
2) Even if the ground truth hand motions are accurate, there may be multiple possible hand motions to generate a given performance audio, none more "correct".
However, I think this may be okay if the dataset is positioned as "how some pianists performed these pieces" rather than "how one *should* perform these pieces".

An uncited research paper related to the above point is RoboPianist (Zakka et al. 2023): https://kzakka.com/robopianist/

I'm not sure why the baseline method included attempts to infer hand motions directly from audio, rather than going through MIDI.  MIDI (including note velocities) seems like it would contain all of the information needed to reconstruct the hand motions, and is *much* more compact than the raw audio.  If this is incorrect, it would still probably be worth mentioning in the paper.

The paper doesn't include a discussion of sustain pedal.  It seems as though the piano transcription method used *does* transcribe pedal events, which seem like they would be necessary to include in the baseline method otherwise much piano audio would be impossible to perform hands-only.  However, since the baseline method goes straight from audio to hand motions, the model is forced to implicitly transcribe sustain pedal in order to determine the hand motions.  Is that correct?

The paper raises a bunch of questions that I am curious about but were not discussed, specifically related to limits of the method and failure modes.  The examples included in the supplementary material are sort of "easy".  What happens if you give the baseline method virtuosic piano audio e.g. the MAESTRO dataset?  What happens if you speed up the audio so that it's no longer humanly playable?  What happens if you play intervals that human hands can't accommodate?

**Questions:**

I don't have questions though I'd probably accept the paper as is.

---

> ### Author Response · Authors · 2024-11-20
> **Response to Reviewer Jc4e**
>
> ### Q1: Physical Verification of Hand Motions
>
> **R:** Thanks for your contructive feedback. Verification of hand motion sequences with their target audio using a physical model would indeed enhance data reliability. In future work, we plan to expand the dataset and incorporate a simulation environment, such as the one proposed by Zakka _et al._ (2023), to simulate piano performances and assess the precision of generated motions. This method facilitates dataset refinement by identifying and rectifying inaccuracies. Additionally, simulated environments could facilitate MIDI generation directly, further enhancing dataset precision.
>
>
> ### Q2: Multiple Possible Hand Motions for a Given Performance
>
> **R:** As noted in our abstract, “PianoMotion10M aims to provide guidance on piano fingering for instructional purposes.” Our method enables **diverse piano performances** for a given audio piece. The diffusion model we developed supports this flexibility by simulating different playing styles under the same audio input, with variations controlled via seed adjustments.
>
> ### Q3: Baseline with Audio vs. MIDI
>
> **R:** MIDI does indeed provide a more direct description of notes and positions, which could simplify the estimation of hand motions. We also conducted experiments on hand motions generated from MIDI, which can be found in R4 of **Global Author Rebuttal**. However, an end-to-end approach from audio to hand motions is generally more practical, which led us to design our baseline model based on this motivation.
> As for the sustain pedal, we agree that it plays a pivotal role in achieving realistic performances.  Currently, our audio-based approach requires the model to implicitly account for pedal use, and we will explore more explicit treatments of pedal events or rhythms in future work.
>
> ### Q4: Exploration of Model Limitations and Failure Modes
>
> **R:** Thank you for these thought-provoking questions. Our baseline model is learned on data within **typical human performance ranges**, while giving challenging cases—such as virtuosic or unnaturally sped-up piano audio—it may struggle to generate realistic hand motions. These cases fall outside the distribution of the training data, making it difficult for the model to handle extreme tempos or hand spans that exceed human limitations. This is an area we recognize as a limitation when dealing with out-of-distribution (OOD) inputs, and future work could explore adjustments for handling such scenarios more robustly. We discuss the failure cases and provide visualization samples in the appendix of the revised submission.

---

### Official Review · Reviewer_RbsM · 2024-11-04

**Soundness:** 3
**Presentation:** 3
**Contribution:** 3
**Rating:** 6
**Confidence:** 4

**Summary:**

This paper presents PianoMotion10M, a large-scale dataset designed to advance hand motion generation in piano performance, featuring 116 hours of piano videos annotated with 10 million hand poses. To complement this dataset, the authors introduce a baseline model that generates realistic and continuous hand motions from piano audio, using a position predictor and a gesture generator. The dataset and model are aimed at improving AI-driven piano instruction, with metrics to assess motion quality, smoothness, and positional accuracy.

**Strengths:**

1. Originality and Dataset Contribution: The PianoMotion10M dataset provides a novel resource, combining piano audio and video data with detailed 3D hand pose annotations across 116 hours of performance. This dataset addresses a significant gap by providing high-resolution, continuous hand motion data, which is essential for realistic AI-driven piano instruction and training, especially in guiding hand positioning and transitions.

2. Comprehensive Benchmark and Evaluation Metrics: The paper introduces a set of detailed evaluation metrics, including Frechet Gesture Distance, Wasserstein Gesture Distance, and Position Distance, to assess motion fidelity, positioning accuracy, and smoothness. This comprehensive approach to benchmarking supports the model’s effectiveness and provides a robust foundation for future research in hand motion generation.

3. Well-defined Baseline Model: The baseline model combines a position predictor with a gesture generator to generate realistic hand motions based on audio input, demonstrating the dataset’s utility. The use of a diffusion probabilistic model for gesture generation adds an innovative approach to capturing continuous motion, enhancing the quality of generated hand movements and reinforcing the practical application of PianoMotion10M in piano instruction.

**Weaknesses:**

1. Supplementary Material and Generation Results: The supplementary material lacks corresponding piano music for the generated hand motions, making it difficult to assess the quality and realism of the generated performance. Including paired audio and visual outputs would provide a more complete evaluation of the model’s effectiveness.

2. Dataset Collection Process:
The collection and preprocessing of data, essential to any dataset creation work, lack clarity. Specifically, the paper does not thoroughly explain the data acquisition process in Section 3.1.
While the authors mention in line 213 that "3,647 candidate videos" were collected, it remains unclear how these videos were queried or selected.
In line 240, the authors state that they "remove videos where hand regions are frequently obstructed or invisible." However, they do not provide a clear definition of what constitutes "frequently obstructed or invisible," making it hard to understand the filtering criteria.
Line 236 mentions manually selecting pure piano music to ensure no vocals or sounds from other instruments, but it’s unclear why an automated audio event detection approach (such as [1]) wasn’t considered for this task.

3. Annotation Quality and Relevance:
The quality of annotations is crucial in generating reliable datasets. In line 251, the authors mention "we transcribe the piano performances into MIDI files by utilizing a state-of-the-art automatic piano transcription (Kong et al., 2021)". However, this method, although effective, was developed in 2021. The authors do not address whether they explored more recent transcription methods (such as [2-5]) or provide experiments comparing different methods to validate their choice.

4. Additionally, while MIDI signals are provided, they are not used in the hand motion generation task. It would be beneficial to validate the quality and relevance of MIDI annotations, potentially by incorporating MIDI data into the baseline model to assess its impact on hand motion generation. This could offer valuable insights for future research.

[1] Kong Q, Cao Y, Iqbal T, et al. Panns: Large-scale pretrained audio neural networks for audio pattern recognition[J]. IEEE/ACM Transactions on Audio, Speech, and Language Processing, 2020, 28: 2880-2894.

[2] Maman B, Bermano A H. Unaligned supervision for automatic music transcription in the wild[C]//International Conference on Machine Learning. PMLR, 2022: 14918-14934.

[3] Gardner J, Simon I, Manilow E, et al. MT3: Multi-task multitrack music transcription[J]. arXiv preprint arXiv:2111.03017, 2021.

[4] Toyama K, Akama T, Ikemiya Y, et al. Automatic piano transcription with hierarchical frequency-time transformer[J]. arXiv preprint arXiv:2307.04305, 2023.

[5] Cheuk K W, Sawata R, Uesaka T, et al. Diffroll: Diffusion-based generative music transcription with unsupervised pretraining capability[C]//ICASSP 2023-2023 IEEE International Conference on Acoustics, Speech and Signal Processing (ICASSP). IEEE, 2023: 1-5.

**Questions:**

Please see Weakness.

---

> ### Author Response · Authors · 2024-11-20
> **Response to Reviewer RbsM**
>
> ### Q1: Outputs with Paired Audio and Visualization
>
> **R:** Thank you for the feedback. Our project webpage includes paired audio along with the corresponding generated hand motions. Though the volume may be somewhat low; we recommend increasing the volume for optimal synchronization and quality assessment. Additionally, some videos in the supplementary material may have audio playback issues due to encoding format incompatibility. We have found that QuickTime may not support the audio format, but using players like ffplay or IINA can resolve this issue.
>
> ### Q2: Clarity of Dataset Collection and Filter
>
> **R:** Thanks for your seggestion. Please refer to our previous response (R2) in **Global Author Rebuttal**. To enhance the clarity of our dataset collection process: we selected videos that best highlight hand movements by prioritizing those shot from a bird’s-eye view to minimize hand occlusion. We reached out to specific content creators for permission and obtained raw videos directly from them. During manual filtering, we excluded videos that did not meet our angle requirements, involved multiple performers or instruments, or contained distracting sounds. Automated audio event detection alone was insufficient for our needs due to potential inaccuracies in filtering out non-piano sounds but also the additional filtering criteria such as viewpoint, hand visibility, and the number of hands were essential for our dataset quality.
>
> ### Q3: Annotation Quality and Recent Transcription Methods
>
> **R:** Thank you for highlighting advancements in music transcription. We initially adopted the method by Kong _et al._ (2021) due to its demonstrated robustness in various scenarios. Also, we recognize that recent methods like MT3 (Gardner _et al._) and DiffRoll (Gardner _et al._) may offer the improved accuracy. To investigate this, we randomly selected 100 audio samples in our dataset and transcribed them into MIDI using various methods. The approaches by Maman _et al._ and Toyama _et al._ lack publicly available inference code, and Diffroll, which relies on diffusion with a fixed sequence length, poses challenges in smoothly stitching MIDI sequences. We evaluated the method proposed by Kong _et al._ against MT3, focusing on **Mel Spectrogram Loss**（L2), **rhythm stability**, and **temporal consistency**, as shown in the table below. Rhythm stability was assessed by analyzing note intervals and tempo patterns in the MIDI files, while temporal consistency was measured using Dynamic Time Warping distance to compare the rhythm of the MIDI with the original audio. Although both methods show similar performance in audio estimation, the approach by Kong _et al._ demonstrates superior stability in rhythm and prosody.
>
> To enhance annotation quality, we manually refined MIDI files with significant discrepancies, ensuring alignment with the video content. Additionally, some creators provided ground-truth MIDI for their performances, and we will acknowledge their contributions in the camera-ready version.
>
> |Model|Mel spectrogram Loss|Rhythm stability|Temporal Consistency|
> |:-:|:-:|:-:|:-:|
> |MT3| **350.17** | 6.72 | 216.9 |
> |Kong _et al._ (2021)| 352.41 | **0.89** | **197.3** |
>
> ### Q4: Hand Motion Generation with MIDI
>
> **R:** Thanks for your thoughtful comments. MIDI data does offer a direct link to hand movements and positions, while we focused on audio-to-pose generation in this version as audio data is generally more accessible than MIDI. More experimental results can be found in the R4 of **Global Author Rebuttal**. Consequently, our baseline model was designed to generate hand movements directly from piano audio, which provides broader applicability and ease of use for this dataset.

---

### Author Response · Authors · 2024-11-20
**Global Author Rebuttal (Part 2/2)**

### MIDI-guided Hand Motion Generation

We conducted a preliminary experiment on MIDI-guided hand motion generation using a similar structure, and replaced Wav2Vec with an MLP as the backbone. As shown in the following table, the experimental results indicate that our network can effectively handle MIDI-to-pose generation as well. MIDI provides more precise hand position estimates, while the pretrained Wav2Vec backbone allows audio input to achieve better performance in motion similarity. This is likely due to the simplicity of the MLP backbone. These findings further confirm that our dataset can support multiple tasks, providing flexibility for various studies. Visualization results have been provided on our project web page.

|Guidance|Backbone|FGD|WGD|PD|Smooth|FID|
|:-:|:-:|:-:|:-:|:-:|:-:|:-:|
|Audio|Wav2Vec2.0 | 0.36 | 0.23 | 0.034 | 0.25 | 3.38 |
|Audio|HuBert | **0.36** | **0.23** | 0.035 | **0.24** | **3.28** |
|MIDI|MLP | 0.39 | 0.25 | **0.031** | 0.31 | 3.85 |

### User Study

Due to limited period of time, a small scale human evaluation was conducted with 10 piano experts who assessed the quality of piano-playing hand movements. Each expert was assigned 36 pairs of anonymized videos, each pair presents two different hand motion sequences synchronized to the same background music. Experts evaluated the hand motions based on the following four criteria:

- **Rhythmic Synchronization:** How well the hand movements align with the rhythm and tempo of the accompanying music, assessing the timing accuracy of hand transitions and finger tapping.
- **Motion Smoothness:** The fluidity of hand and finger movements throughout the sequence, with attention to the continuity and natural flow of transitions.
- **Motion Realism:** The degree to which the hand motions resemble natural, human-like movements, emphasizing the believability of the generated performance.
- **Motion Variety:** Observing the range of techniques and transitions, specifically the diversity and dynamics in moving from one hand position to another, indicating realistic variability and adaptive finger movement.

The evaluation compared the following methods: EmoTalk, LiveSpeaker, ours large-HuBERT-Mamba, and large-HuBERT-Transformer. As presented in the table below, end-to-end methods like EmoTalk exhibit  limitations, particularly with hand and finger motions, which are less pronounced. This impacts alignment with the rhythm and reduces motion diversity. Additionally, evaluation metric FID=4.65 also indicates a significant distributional gap between hand gestures generated by EmoTalk and the hand motion distribution in real-world. The diffusion-based method LivelySpeaker shows a significant improvement in motion diversity, even slightly surpassing our Large-HuBERT-Mamba, suggesting a strong grasp of short-term actions. However, it performs less well in aligning motions with rhythm over longer sequences. Quantitative metrics substantiate this observation. WGD of LivelySpeaker approximates our baseline, whereas its FGD and FID are significantly reduced. Additionally, among our two network choices, the transformer-based approach achieves higher motion diversity with a higher FID than the Mamba-based model. These findings substantiate the efficacy of our proposed metrics, enabling cost-effective model performance assessment compared to more resource-intensive human evaluations.

|Model|Rhythmic Synchronization|Motion Smoothness|Motion Realism|Motion Variety|
|:-:|:-:|:-:|:-:|:-:|
|EmoTalk | 3% | 42% | 36% | 26% |
|LivelySpeaker| 56% | 52% | 49% | 46% |
|Large-HuBERT-Mamba| 65% | **63%** | 57% | 45% |
|Large-HuBERT-Transformer| **76%** | 57% | **58%** | **83%** |

---

### Author Response · Authors · 2024-11-20
**Global Author Rebuttal (Part 1/2)**

We sincerely thank all the reviewers for their valuable time and constructive comments. We are encouraged that the reviewers appreciate the **enough novelty** (RbsM, Jc4e, ik5L), **certain influence** (RbsM, Jc4e, wnCC, ik5L), **well-organized writing** (RbsM, Jc4e), **extensive dataset** (wnCC, ik5L) and **excellent experimental performance** (RbsM, wnCC). All suggestions are seriously considered. Moreover, we have carefully revised the manuscript according to the comments. In this section, we address the commonly raised concerns. Then we reply each reviewer point by point in individual comments.

### Motivation and Novelty.
The paper introduces PianoMotion10M, a novel large-scale piano-motion dataset for hand motion generation and music synthesis, as acknowledged by reviewer comments. **Reviewer RbsM** finds the task and dataset for piano-hand motion generation essential for realistic AI-driven piano instruction and training. **Reviewer Jc4e** notes that it provides valuable resources for the research community. **Reviewer ik5L** mentions that this represents a novel application domain with the potential for widespread impact across multiple tasks.

### More Details of Data Collection

According to the suggestions from **reviewer ik5L** and **RbsM**, we have incorporated a comprehensive description on the dataset collection process along with a thorough discussion in the revised manuscript.

**More details of data filter:** To accurately capture hand movements, we primarily collected bird’s-eye view videos in collaboration with content creators. We manually filtered out videos that did not meet our criteria, such as those featuring multiple performers, additional instruments, or background noise. Videos in which hands were visible in less than 80% of frames were excluded to prevent occlusions. Although automated audio event detection was considered, manual selection was ultimately employed to ensure higher accuracy by filtering out non-compliant videos.

**More details of MIDI adjustment:** In our work, we defined “high discrepancy” as a deviation that exceeds a threshold of 15% in terms of note timing or velocity when compared to a reference dataset or ground truth. Specifically, we replayed the MIDI files and compare them directly to the original music tracks. Files were adjusted if they exceeded thresholds such as timing differences over 30 ms, dynamic variations beyond 10%, or noticeable pitch mismatches that alter melody or harmony. These criteria help the MIDI files closely match the original music, enhancing the dataset’s reliability as a robust benchmark.

**More details of smooth filter:** We applied a Savitzky-Golay filter with an order of 3 with a window size of 11 to smooth the hand pose data extracted by HaMeR, which is based on single-frame predictions. This filter was chosen due to its ability to preserve critial motion details while effectively reducing high-frequency jitter, which would otherwise be present without smoothing.
The Savitzky-Golay filter delivers better results compared to Butterworth filter in terms of noise separation, artifacts and baseline drifts, making it more suitable for our task. Without the filter, the data would exhibit noticeable jitter, making the motion sequences less stable and realistic for analysis and model training.


### Ethical Concerns and Copyright Considerations

During the process of collecting dataset, we carefully selected the high-quality piano performance videos with a bird’s-eye view on video sharing website (Bilibili) and obtained official permissions from the respective creators to use their videos in research, and accessed the original content. Those videos from creators who either did not grant permission or did not respond to our requests have not been included in the dataset. We informed the creators that we would extract hand poses and audios from the videos in order to create a publicly available, non-commercial dataset.
We clarified that their personal likenesses would not be used. The dataset consists of hand pose, audio and corresponding MIDI file but does not contain hand images. It also included metadata, such as the video URL, to reference the original video. In the camera-ready version, we will acknowledge all the creators for their contributions to building our dataset.
Moreover, if creators wish to have their videos removed, they will be excluded from subsequent dataset updates. We ensure that all data usage strictly complies with the permissions granted by the creators.

---

### Comment · Area_Chair_9qot · 2024-11-24
**Public discussion phase ending soon**

Dear reviewers,

Thank you for your diligent work on the reviews. Currently the paper has a very uniform score: 6, and the authors have responded to every single one of the reviews.

All reviewers: did the authors' rebuttals and other reviews affect your score? Please respond before the 26 November to let the authors have time to respond if you need any clarification. Thank you!

Your AC

---

### Author Response · Authors · 2024-12-02

Dear reviewers,

We sincerely thank the reviewers for your insightful comments and valuable suggestions, which contributed to improving the quality of our work. We have carefully addressed each point and highlighted the corresponding changes in blue throughout the revised manuscript. Your thoughtful feedback has been instrumental in refining our paper, and we deeply appreciate your time and effort.

PianoMotion10M authors

---

### Meta-Review · Area_Chair_9qot · 2024-12-23

**Metareview:**

This paper introduces PianoMotion10M, a new dataset of piano performances with video, audio, MIDI, and hand motion data.  The dataset contains over 116 hours of piano videos annotated with 10 million hand poses. The authors also introduce a baseline model that generates realistic and continuous hand motions from piano audio, using a position predictor and a gesture generator. The dataset and model are aimed at improving AI-driven piano instruction, with metrics to assess motion quality, smoothness, and positional accuracy.

Strengths:
Provides a large and unique dataset for piano performance research.
Includes accurate 3D hand pose annotations.
Offers a baseline model for hand motion generation (position predictor with a gesture generator, using a U-net and diffusion model).
Defines clear evaluation metrics (including Frechet Gesture Distance, Wasserstein Gesture Distance, and Position Distance).

Weaknesses:
Lacks verification that the hand motions can reproduce the audio (addressed with a user study in the rebuttals).
Baseline model doesn't utilize MIDI information (addressed in rebuttals with new experiments).
Limited diversity in piano positions and playing styles.
Could benefit from more detailed explanations of data collection and annotation choices (addressed in rebuttals).

**Additional Comments On Reviewer Discussion:**

Only one reviewer engaged with the authors' rebuttal and raised their score from 6 to 8.

---

### Decision · Program_Chairs · 2025-01-22

Accept (Spotlight)